



# Responses of soil water storage and crop water use efficiency to changing climatic conditions: A lysimeter-based space-for-time approach

Jannis Groh[1, 2], Jan Vanderborght[2], Thomas Pütz[2], Hans-Jörg Vogel[3], Ralf Gründling[3], Holger Rupp[3], Mehdi Rahmati[4], Michael Sommer[5, 6], Harry Vereecken[2], Horst H. Gerke[1]

[1]Research Area 1 "Landscape Functioning", Working Group "Hydropedology", Leibniz Centre for Agricultural Landscape Research (ZALF), Müncheberg, 15374, Germany
[2]Institute of Bio- and Geoscience IBG-3: Agrosphere, Forschungszentrum Jülich GmbH, Jülich, 52425, Germany
[3]Department of Soil Physics, Helmholtz Centre for Environmental Research — UFZ, Halle (Saale), 06120, Germany
[4]Department of Soil Science and Engineering, Faculty of Agriculture, University of Maragheh, Maragheh, Iran
[5]Research Area 1 "Landscape Functioning", Working Group "Landscape Pedology", Leibniz Centre for Agricultural Landscape Research (ZALF), Müncheberg, 15374, Germany
[6]Institute of Environmental Science and Geography, University of Potsdam, Potsdam, 14476, Germany

*Correspondence to*: Jannis Groh (groh@zalf.de or j.groh@fz-juelich.de)

**Abstract.** Future crop production will be affected by climatic changes. In several regions, the projected changes in total rainfall and seasonal rainfall patterns will lead to lower soil water storage (SWS) which in turn affects crop water uptake, crop yield, water use efficiency, grain quality and groundwater recharge. Effects of climate change on those variables depend on the soil properties and were often estimated based on model simulations. The objective of this study was to investigate the response of key variables in four different soils and for two different climates in Germany with different aridity index: 1.09 for the wetter (range: 0.82 to 1.29) and 1.57 for the drier climate (range: 1.19 to 1.77), by using high-precision weighable lysimeters. According to a "space-for-time" concept, intact soil monoliths that were moved to sites with contrasting climatic conditions have been monitored from April 2011 until December 2018.

Evapotranspiration was lower for the same soil under the relatively drier climate whereas crop yield was significantly higher, without affecting grain quality. Especially 'non-productive' water losses (evapotranspiration out of the main growing period) were lower which led to a more efficient crop water use in the drier climate. A characteristic decrease of the SWS for soils with a finer texture was observed after a longer drought period under a drier climate. The reduced SWS after the drought remained until the end of the observation period which demonstrates carry-over of drought from one growing season to another and the overall long term effects of single drought events. In the relatively drier climate, water flow at the soil profile bottom showed a small net upward flux over the entire monitoring period as compared to downward fluxes (ground water recharge) or drainage in the relatively wetter climate and larger recharge rates in the coarser- as compared to finer-textured soils. The large variability of recharge from year to year and the long lasting effects of drought periods on SWS imply that long term monitoring of soil water balance components is necessary to obtain representative estimates. Results confirmed a



more efficient crop water use under less optimal soil moisture conditions. Long-term effects of changing climatic conditions

on the SWS and ecosystem productivity should be considered when trying to develop adaptation strategies in the agricultural

sector.

# 1 Introduction

The amount of water stored within the root zone of the soil and the vadose zone is a central and characteristic component of

terrestrial ecosystems. Soil water storage (SWS) is important for provisioning (e.g., crop production, water balance, and

plant available nutrients) as well as regulating and supporting ecosystem services (e.g. water, nutrients, climate, flood,

drought; Adhikari and Hartemink, 2016; Vereecken et al., 2016). The SWS capacity (SWSC) depends on soil texture,

organic matter content, bulk density, and soil structure and is related to the effective field capacity, which can be derived

from the soil water retention function (Vereecken et al., 2010). The knowledge on magnitude and temporal variation of the

SWS is essential for understanding ecological and hydrological processes and to manage ecosystems (Cao et al., 2018).

Climate change will modify the temporal availability of soil water, increase the frequency and duration of droughts, affecting

the quantity and quality of aquifer recharge and might affect crop production. Thus future ecosystem productivity (e.g. crop

yield) is expected to respond to changes in weather (short-term) and climate (long-term), because it will alter the crop water

balance components, such as SWS, evapotranspiration (ET) and drainage (Yang et al., 2016). How to produce more crop

yield with less water is a major challenge in agriculture, because i) water is a limiting factor for crop production in many

regions of the world, and ii) predictions of future climate indicate an increasing water limitation for crop production caused

by reduced rainfall and changing seasonal rainfall distribution (Lobell and Gourdji, 2012).

Several studies have been conducted to investigate the impact of global climate change on crop water balance components

(Sebastiá, 2007; Wu et al., 2015) and crop or grain yield (Ewert et al., 2002; Zhao et al., 2016; Schauberger et al., 2017;

Asseng et al., 2019). Understanding the impact of weather signals on the agricultural productivity is of crucial importance

for managing future crop production, since variations in weather conditions could explain much of the yield variability

(Frieler et al., 2017). Temperature rise and changing seasonal rainfall patterns could alter the probability of droughts and

affect freshwater resources (Gudmundsson and Seneviratne, 2016; Gudmundsson et al., 2017). Negative impacts of rising

temperature on the yield of major crops at the global scale (Asseng et al., 2014; Zhao et al., 2017) are highlighting the

potential vulnerability of agricultural productivity to climate change. Schauberger et al. (2017) showed a consistent negative

response of US crops under rainfed conditions being mainly related to water stress induced by higher temperatures. In

addition to the direct effects of a temperature rise, an elevated atmospheric $CO_2$-concentration, and changes in rainfall

amounts on crop yield (Ewert et al., 2002; Asseng et al., 2014; Gammans et al., 2017; Scheelbeek et al., 2018), the higher

temperatures could affect crop yields indirectly. Indirect effects caused by increasing the atmospheric water demand, limiting

ET due to water stress and reducing the SWS, could in turn lead to a decrease in crop yield (Zhao et al., 2016; Zhao et al.,





2017). Thus investigating the response of crop water balance components and yield to climate change is important to develop suitable adaptation and mitigation strategies (Albert et al., 2017; Rogers et al., 2017).

Previous studies reported estimates of crop water balance components and crop yield mostly based on either manipulative experiments or observational studies to predict the ecological response of crops to climate change (Yuan et al., 2017). Wu et al. (2015) showed that the inter-annual variation of the SWS at northern middle and high latitudes increased under a warmer

climate with higher values during the wetter and lower values of the SWS during the drier season. In this case, the frequency of water logging events or soil crack formation will increase and probably alter soil properties such as macroporosity and SWSC and thus affect vadose zone hydrology at different scales (Robinson et al., 2016; Hirmas et al., 2018). Robinson et al. (2016) showed for a manipulative long-term experiment that intense summer droughts altered the soil water retention characteristic and lowered the SWSC.

Nevertheless, current knowledge on changes of SWS are still limited mostly to the analysis of soil moisture observations related to restricted soil volumes and soil moisture ranges (Mei et al., 2019; Yost et al., 2019). As an alternative method, weighable lysimeters allow the direct observation of SWS by monitoring the temporal changes of the total soil mass in mostly cylindrical containers. However, the use of weighable lysimeters was often limited in the past to the quantitative determination of the water balance components of precipitation ($P$), evapotranspiration (ET), and subsurface inflow ($Q_{in}$) and

outflow ($Q_{out}$; e.g. drainage); the change of SWS was obtained as residual of the water balance components (e.g. Herbrich et al., 2017; Groh et al., 2018b). This approach accumulated all possible errors introduced by other components into the SWS, causing a relatively low precision. The direct derivation of SWS from lysimeter mass changes could provide a new perspective on the use of lysimeter data as an additional model calibration variable and for lysimeters that are large enough to fully capture the complete soil profile with the relevant soil horizons and intact soil structures to be representative for the

pedon scale.

Crop water use efficiency (WUE), being the ratio between grain yield or total biomass and the water lost to the atmosphere by ET, is one of the possible ways to quantify the impact of changes in the environmental conditions and of management decisions (e.g. irrigation) on agricultural productivity. The WUE provides insights to better manage and understand the productivity and ecological functioning of agricultural ecosystems (Zhang et al., 2015). The prevailing general hypothesis

for WUE is that plant productivity increases with increasing water use (ET; Hatfield and Dold, 2019), which implies that WUE efficiency is a linear function of the water used by a crop to produce grain yield or the total above ground biomass. But several studies have shown that crop WUE was negatively correlated with annual rainfall and plants achieved their maximum crop WUE under less favourable soil water availability (Zhang et al., 2010; Ponce-Campos et al., 2013; Xiao et al., 2013; Zhang et al., 2015). The last statement might imply that plants are able to adapt their water use during drought

conditions by improving their WUE or that there is simply less non-productive water losses by evaporation. Nevertheless, temperature above a certain threshold (extremely high temperature) especially during the reproductive period (Gourdji et al., 2013) or due to drought and heat stress reduce yield. However, such investigations are often focused on one specific environmental variable (e.g. $P$ or temperature) in manipulation experiments. This basically ignores joint effects of several



climate variables on the crop WUE in climate impact research studies. The impact of altered climatic conditions on different
agricultural ecosystems within manipulative experiments has not been thoroughly studied yet; due to problems to either
realistically manipulate the climatic conditions at a specific site or to move an intact soil to another site with contrasting
climatic conditions.

Here, we hypothesize that WUE will not increase for drier climate; because a change in plant productivity will
simultaneously alter the water use (ET) and thus describe WUE as a linear function between both variables. In addition we
wanted to test if observed lysimeter mass changes can be used to monitor the long-term change of SWS, which might be in
addition to water flux observation a useful dataset for the calibration of vadose zone models. We used observations from a
German soil-climate crossed factorial experiment (TERENO-SOILCan; Pütz et al., 2016). The lysimeter network of
TERENO-SOILCan has been initiated to assess effects of climatic changes on arable and grassland soil ecosystems
including the water balance components (ET, SWS, net drainage) and crop characteristics including yield, yield quality and
WUE. As part of this project, arable-land lysimeters filled with four different soils were transferred within and between
TERENO observatories (space-for-time; see details in Pütz et al., 2016) to expose soils from originals sites to other climatic
conditions. The space-for-time approach means that soils are translocated in space instead of waiting at the same location for
changes in climatic conditions in time. The concept initially intended to evaluate the impact of climate on agricultural
ecosystems (Pütz et al., 2016). It represents basically a crossed soil type and climate experimental setup that could allow
quantifying changes in the soil water balance and the crop production as response to imposed variations in climatic
conditions. Results from this experimental setup can primarily be used to evaluate models that predict changes in response to
possible future climatic conditions.

Our objectives were: i) to develop an approach to obtain time series of changes in SWS directly from lysimeter data , ii) to
determine the other soil water balance components ($P$, ET, inflow and drainage) of soils each exposed to two different
climates, iii) to compare the net flux (inflow and drainage)/SWS dynamics for the same soils in relatively dry and wet
climates and iv) to test the hypothesis that WUE of crops remains constant under changing climatic conditions in these
crossed soil type and climate experiment. The analysis was based on lysimeter data from April 2011 until December 2017.

## 2 Material and Methods

### 2.1 Site descriptions

The study was conducted at the experimental field sites Selhausen (50°52´7´´N, 6°26´58´´E) and Bad Lauchstädt
(51°23´37´´N, 11°52´41´´E), which are part of the Eifel/Lower Rhine Valley and the Harz/Central German Lowland
Observatory of TERENO in Germany (Wollschläger et al., 2016; Bogena et al., 2018), respectively. The TERENO-
SOILCan lysimeter network was established at several experimental stations across a rainfall and temperature gradient.
Local excavated lysimeters (i.e. intact soil monoliths) were transferred between the stations to subject them to different



climate regimes so as to generate a crossed soil-climate setup according to the space for time approach. For this study, we considered all arable-land lysimeter at the central sites Bad Lauchstädt and Selhausen of the TERENO-SOILCan lysimeter network. Each central experimental site contains three replicates of soils from different locations: Bad Lauchstädt (BL; Haplic Chernozems, loess), Dedelow (Dd; Calcic Luvisols and Haplic Luvisols, glacial till), Sauerbach (Sb; Colluvic

Regosols; colluvial deposits), and Selhausen (Se; Haplic Luvisols, loess) allowing to investigate the response of the corresponding soil type under different climates. Further information on the transfer of soil monoliths from the TERENO-observatories to the central sites can be taken from Table A1 (see appendix). The transferred eroded Luvisol soil monoliths from Dedelow have a varying soil depth to the clay illuviation horizon ($B_t$) and to the marly, illitic glacial till ($C$-horizon). They represent part of the erosion gradient typically observed in agricultural landscapes of hummocky ground moraines

(Sommer et al., 2008; Rieckh et al., 2012; Herbrich et al., 2017). Detailed information about the lysimeter design and general experimental-set up of TERENO-SOILCan can be found in Pütz et al. (2016). The climatic conditions of the central sites from 1 January 2012 to 31 December 2017 (complete years) are shown in Fig. 1 according to Walter and Lieth (1967). Although the patterns in average monthly temperature values are relatively similar at both sites (Fig. 1), a more pronounced amplitude of the temperature variations over the year could be found in Bad Lauchstädt (representing a more continental

climate) as compared to the more temperate and humid climate (sub-oceanic or sub-Atlantic) in Selhausen (Fig. 1). The average annual grass reference evapotranspiration ($ET_0$) calculated with the FAO56 Penman-Monteith method (Allen et al., 2006) is slightly higher at Bad Lauchstädt (710 mm) than at Selhausen (694 mm). Larger differences are shown in the annual rainfall and the rainfall distribution over the year (Fig. 1). The lower annual $P$ in Bad Lauchstädt (458 mm) than in Selhausen (644 mm) corresponds with a higher aridity index (AI = $ET_0 \, P^{-1}$, see data repository) of 1.57 for Bad Lauchstädt

than for Selhausen (1.09). The rainfall distribution over the year was more uniform in Selhausen whereas the probability of relatively dry periods in spring (April) and late summer (September) was higher in Bad Lauchstädt. Thus, the climatic conditions at the SOILCan experimental sites can be defined as drier for Bad Lauchstädt and wetter at Selhausen, which corresponds well to long-term weather station data reported by Groh et al. (2016) for the period from 1981 to 2010.





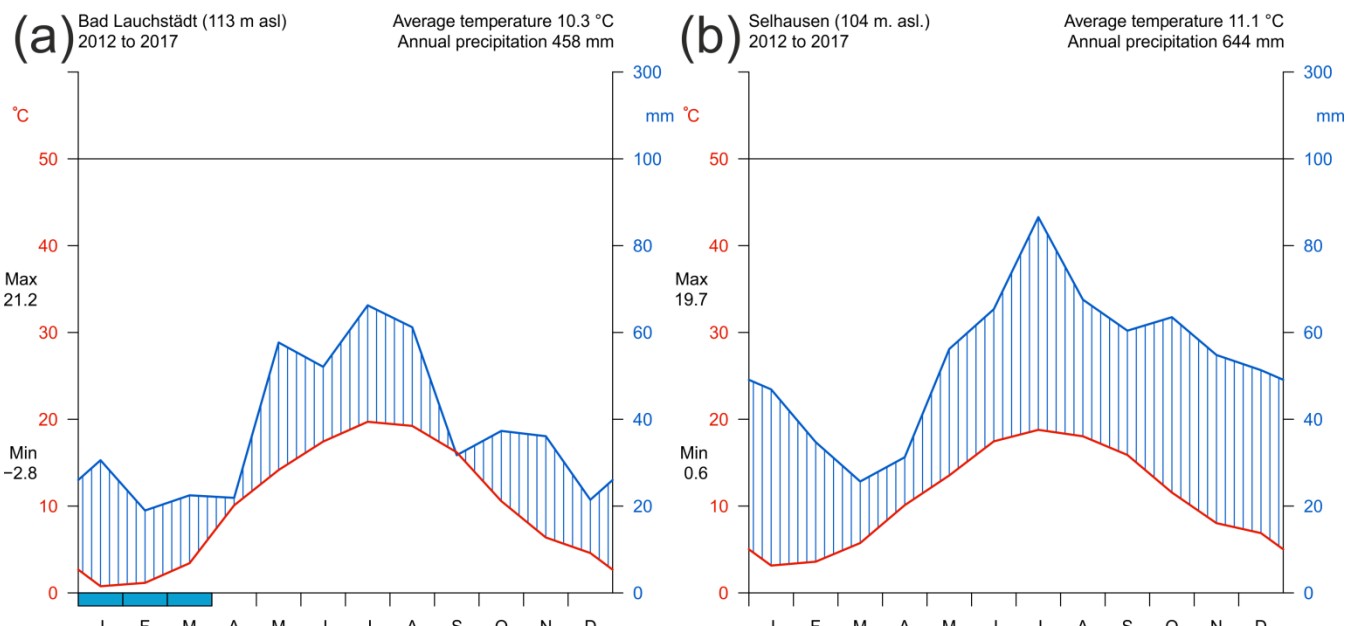

**Figure 1:** Climate diagrams according to Walter and Lieth (1967) for Bad Lauchstädt (a), and Selhausen (b) for 2012 to 2017. Data were obtained from the SOILCan weather stations at Selhausen and a climate station at Bad Lauchstädt above sea level (asl.). The blue bars at the bottom of subplot a) indicate months were frost is likely to occur.

## 2.2 Soil water storage (SWS)

Monthly changes in SWS (ΔSWS) were calculated from lysimeter observations as:

$$\Delta SWS = \Delta W + \Delta L_{yscor} \tag{1}$$

where $\Delta W$ is the monthly lysimeter mass change, and $\Delta L_{yscor}$ corresponds to mass changes by maintenance, harvesting, or other disturbances that occur accidently (e.g. erroneous load cells) or naturally (e.g., animals). The variable $\Delta W$ was directly obtained by analysing lysimeter mass data (average value: 12°AM until 2°AM) defined as:

$$\Delta W = W_{i+1} - W_i \tag{2}$$

where $W$ is the lysimeter mass at the beginning of month $i$. The variable $\Delta L_{yscor}$ was determined from monthly changes of lysimeter mass during maintenance work. Less than 0.6 % of ΔSWS values could not be calculated, because lysimeter mass data at the beginning of the corresponding month were missing. A linear regression model obtained for the entire time series between ΔSWS of the soils was used for interpolation to fill the gaps. This was first based on ΔSWS from surrounding lysimeters of the same soil type and if not available, then the average values of ΔSWS obtained from all available lysimeters at the respective station were used.

## 2.3 Crop water use efficiency (WUE), grain yield and yield quality

In total 12 arable land lysimeters (three replicates of four soil types) with a surface area of 1 m$^2$ and a depth of 1.5 m were embedded within larger fields at the respective central experimental site at Selhausen (250 m²) and Bad Lauchstädt (720 m²).





The same crops were grown and identical tillage and crop management procedures were carried out at both sites and in the

field around the lysimeters. The lysimeters were cultivated with peas (Pisum sativum L.; cultivar: Mascara), winter barley

(Hordeum vulgare L.; cultivar: Lomerit), winter canola (Brassica napus L.; cultivar: Adriana), oat (avena sativa L.; cultivar:

Max G), winter wheat (Triticum aestivum L.; cultivar: Glaucus), winter barley (Hordeum vulgare L., cultivar: Antonella)

and winter rye (Secale cereal L.; cultivar: SU Santini), whereas the applications of seasonal plant protection, crop growth

regulators and nitrogen-fertilizer (see appendix Table A2) have been adapted to local farmer conditions at the respective

experimental site. Dry mass of the yield and plant residual matter were gravimetrically determined with a precision balance

(Selhausen: EMS 6K0.1, KERN, Balingen-Frommern, Germany; Bad Lauchstädt: LC 6200 D, Satorius, Göttingen,

Germany) after drying at 75°C for 24 hours (Bad Lauchstädt) and at 60°C for >24 hours (Selhausen; until reaching a

constant weight). The determination of total nitrogen of the dry yield and plant residual material was obtained with an

elementary analyser (VarioelCube, elementar, Langenselbold, Germany).

The following Eq. (3) was used to calculate the crop WUE (kg m⁻³):

$$\text{WUE} = \frac{Y}{\text{ET}} \qquad\qquad\qquad (3)$$

where $Y$ is the grain yield (kg m⁻²), and ET (m³ m⁻²) is the measure of the consumed water during the growing season of the

corresponding crop (Katerji et al., 2008). The growing periods of the crops were defined as the time between sowing and

harvest (see appendix Table A2). The required ET during the growing season was estimated based on the monthly water

balance equation and observed precipitation ($P$) in mm per month as:

$$\text{ET} = P - \Delta\text{SWS} - Q_{net} - \Delta L_{ysvol} \qquad\qquad\qquad (4)$$

where $Q_{net}$ is the monthly sum of net water flux across the lysimeter bottom ($Q_{net} > 0$: drainage; $Q_{net} < 0$: capillary rise) and

$\Delta L_{ysvol}$ is mass change determined from monthly soil water sampling volume. $P$ was measured with a tipping bucket rain

gauge (15189, Lambrecht, Göttingen, Germany) at Bad Lauchstädt (experimental station Bad Lauchstädt), and with a

weighing rain gauge (Ott Pluvio2 L, Ott, Kempten, Germany) at Selhausen (Se_BDK_002). Data of the latter station is

available at TERENO data portal (http://teodoor.icg.kfa-juelich.de/ddp/index.jsp). The Ott rain gauge was installed in April

2013; data before April 2013 was estimated by linear regression models and $P$ data from surrounding climate stations of the

TERENO data portal (station names: SE_BDK_002; RU_BCK_003; RU_K_001; ME_BCK_001), which can be used to

interpolate between the given data points. We used the R software (R-Core-Team, 2016) and the function lm of the package

stats (R-Core-Team, 2016) to set-up linear regressions. The coefficient of determination ($R^2$) was used to determine the

goodness of fit of the linear regression. A stepwise gap-filling approach was used to gap-fill missing $P$ data after April 2013,

which consisted of an analysis of data from other meteorological stations that were operating and missing values, were filled

based on the observation which had the highest $R^2$. Monthly $Q_{net}$ values were obtained from mass changes of the leachate

from the lysimeters, collected with a weighable reservoir tank. The lysimeter bottom boundary pressure head condition was

imposed by a pumping mechanism, which enabled either outflow or inflow according to differences in pressure head values

at 1.4 m depth between lysimeter and surrounding field soil. This control of the bottom boundary allowed imitating the





upward and downward water fluxes and representation of ET processes in lysimeters (Groh et al., 2016) more realistically and comparable to the intact soil profile. More technical details can be found in Pütz et al. (2016). Missing data in the time series of $Q_{net}$ were filled for small gaps of about one minute by linear interpolation and for gaps between >1 and 10 minutes by using a moving average with a window width of 30 minutes. Larger gaps in the time series were filled by average water flux values from other lysimeters of the same soil type. Nearly 5% of monthly ET values were found not plausible perhaps due to water loss by leaking during periods with water-saturated conditions at the lysimeter bottom. These conditions occurred mainly in winter, when monthly ET fluxes were in general relatively low as compared to summer conditions, so that potential error was low and easily detectable. A linear regression based on either single or average ET values from other non-affected lysimeters with similar soils were used for interpolation to fill the gaps. Detailed information on the monthly water balance data and missing data can be taken from the TERENO data portal (see section Data availability).

## 3 Results and Discussion

### 3.1 Soil water storage change

For the observation period (April 2011- January 2018), evapotranspiration (ET) and cumulative soil water storage change (ΔSWS) differed at both stations, Selhausen and Bad Lauchstädt, in amount and temporal development between transferred soils and those from the original site (Fig. 2). Larger deviations in ΔSWS between origin and transferred soils were visible for the crop winter canola after date of harvest in summer 2013 (soils from BL, Sb, and Se Fig. 2b, 2d, 2h) and winter barley 2016 (all soils). Largest depletions of SWS during the entire observation period could be observed for all soils during the spring-summer period (March and July) in 2015. At Bad Lauchstädt, the aridity index (AI = $ET_0 P^{-1}$) of 2.7 for March-July 2015 was larger as compared to the average AI value of 2.0 calculated for all March and July periods between 2012 and 2017. Also the value of the AI for Selhausen was with 2.0 slightly larger as compared to the average AI value of 1.6 for all March-July periods. The SWS depletion in 2015 was larger at both sites for soils from Bad Lauchstädt (BL; Fig. 2b) and Sauerbach (Sb; Fig. 2d) as compared to that of the other two soils from Dedelow (Dd; Fig. 2f) and Selhausen (Se; Fig. 2h). The Sb and BL soils were strongly desiccated by the winter wheat crop in 2015, which can be seen from ET June 2015 for BL and Sb of about 125 - 175 mm/month (Figs. 2a and 2c) was larger than for Dd and Se soils of about 100 - 125 mm/month (Figs. 2e and 2g) even for the soils exposed to the drier climate in Bad Lauchstädt. For the BL (Fig. 2b) and Sb (Fig. 2d) soils, the amount of rainfall after the growing season of 327 mm (August 2015 - April 2016) in Bad Lauchstädt was not sufficient to compensate for ET and drainage such that the soil profile did not return to a SWS capacity (i.e., typical spring moisture) at the end of the winter period characterized by a value close to 0 of the cumulative ΔSWS. The soil moisture deficit from 2015 was carried over to the growing seasons of 2016 and even of 2017. For the Dd and Se soils (Figs. 2f and 2h, the SWS deficit during the 2015 growing season under the climate of Bad Lauchstädt was less and the amount of precipitation after the growing season was sufficient for the soils to return to a typical SWS value although this value was reached later and not before the next spring. The AI of 1.77 at BL in 2015 (January-December) was considerably higher than





the average AI for the 5 year period at BL (1.57). For the same year 2015, the AI was 1.13 at Se, and thus only slightly higher than the 5-years average AI-value of 1.09. For all soils in Se (blue lines in Figs 2b, 2d, 2f, 2h), the amount of precipitation after the growing season of 501 mm of 2015 (August 2015 - April 2016) was sufficient for the lysimeters to return to their 'typical' SWS value at the end of the winter. These results indicate soil type dependent changes in SWS during drought periods. The annual carry-over of soil moisture deficits demonstrates the vulnerability towards drought risks

even for finer-textured soils, despite having an overall larger SWSC than coarser-textured soils. The observed stronger depletion of soil water corresponds with soil drying reports from larger scale observations on the occurrence of a severe drought during the summer 2015, where effects of the drought has been observed from a climatological (Ionita et al., 2016) and hydrological (Laaha et al., 2016) perspective. The carry-over of soil moisture deficits to the time after the drought at the local scale in Bad Lauchstädt agrees well with the results from Laaha et al. (2016), which showed for several stations in

Europe that soil water storage (catchment scale) at the end of the study period (November 2015) has not totally recovered from the summer drought in 2015.

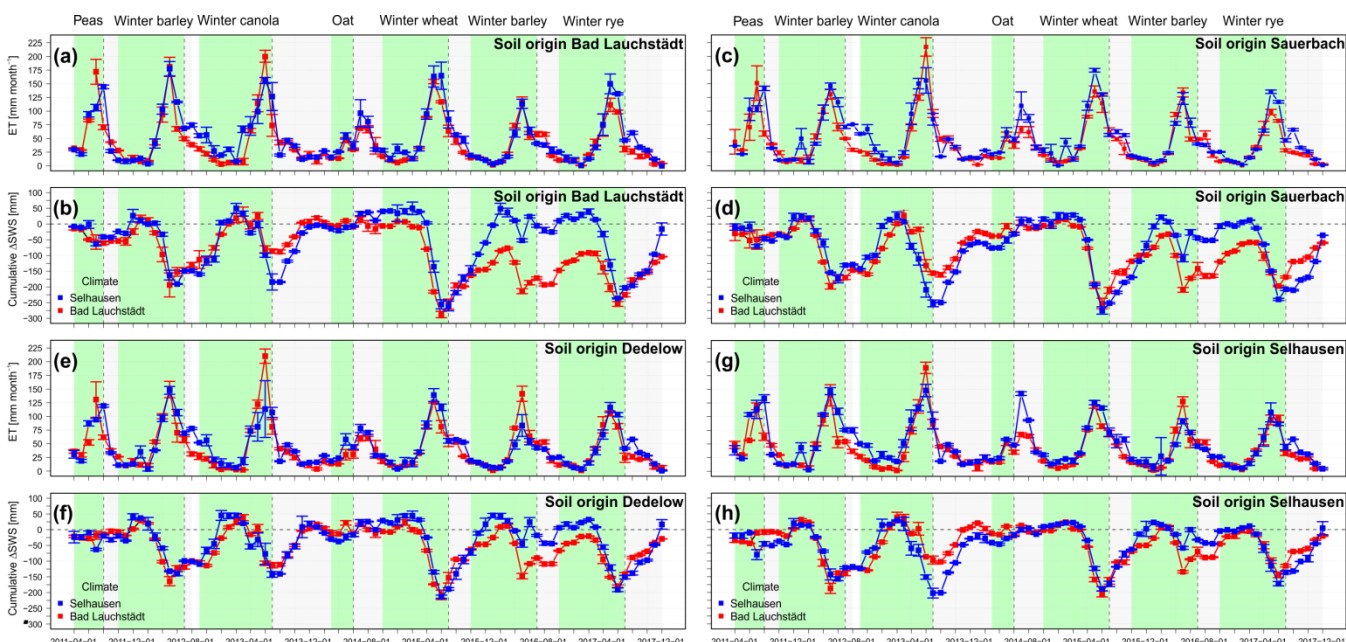

**Figure 2:** Monthly evapotranspiration (ET) and cumulative monthly changes in soil water storage (ΔSWS) from April 2011 until January
2018 at the lysimeter stations in Selhausen and in Bad Lauchstädt for soils from Bad Lauchstädt (a, b), Sauerbach (c, d), Dedelow (e, f), and Selhausen (g, h); mean values (dots) and standard deviations (error bars) are from 3 individual lysimeter monoliths of each soil. The background colour corresponds with the cropping periods at the TERENO-SOILCan lysimeters: bare soil (white) and crops (green).

Furthermore changing climatic conditions and a more frequent occurrence of drought could alter the SWSC because of the
increasingly unavailable pore spaces due to different sources, including physical e.g. swelling and shrinking processes (te Brake et al., 2013; Herbrich and Gerke, 2017), biological e.g. vegetation induced soil desiccation that enhanced soil cracking





(Robinson et al., 2016), biochemical e.g. enhanced organic matter mineralization, due to increasingly oxidation of the organic horizons during dry periods (Robinson et al., 2016), which will consequently result in a degradation of organic soil structure, or change in the soil wettability (Ellerbrock et al., 2005).


## 3.2 Net drainage

The water fluxes across the suction rake system at the lysimeter bottom in 1.5 m depth were cumulated to monthly net drainage fluxes ($Q_{Mnet}$). The time series' of $Q_{Mnet}$ for all soils at Se, the site with relatively wet climate, were in general directed downward during the winter months and upward (capillary rise) during spring and summer (Fig. 3). However, the

magnitude of monthly fluxes $Q_{Mnet}$ differed between the soil types (e.g. soils in Se for 2012 or 2013 see Fig.3); $Q_{Mnet}$ for lysimeters with the coarser-textured soils from Dd (Fig. 3c) was mostly larger (e.g., drainage during bare fallow 2014) than for those with the finer-textured soils from BL (Fig. 3a), Sb (Fig. 3b), and Se (Fig. 3d). For the same soils under the relatively dry climate in BL, time series' of $Q_{Mnet}$ were rather similar, with the largest values of upward fluxes for the soil from Dd (Fig. 3c). The magnitude of $Q_{Mnet}$ for soils under BL climate was mostly smaller for drainage and larger for upward

directed fluxes as compared to the $Q_{Mnet}$ values for the soils under the wet climate in Selhausen.

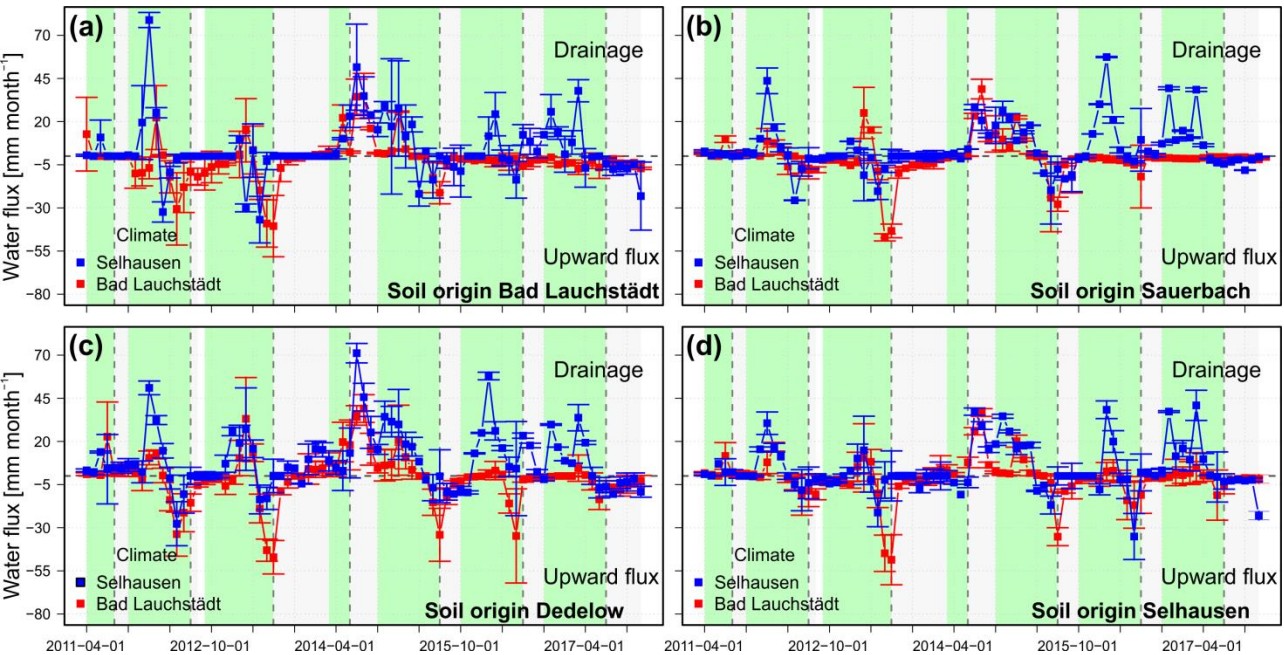

**Figure 3:** Monthly net water fluxes across the lysimeter bottom in 1.5 m soil depth from April 2011 until January 2018 at the stations Selhausen and Bad Lauchstädt for soils from a) Bad Lauchstädt, b) Sauerbach, c) Dedelow, and d) Selhausen; mean values (dots) and
standard deviation (error bars) . Positive values are defined to drainage and negative values to upward direct water flux from capillary rise. Error bars indicate the variability of storage changes between individual lysimeters of each soil group. The background colour corresponds to different crops lysimeter cover types: bare soil (white) and different crops (green).





The $Q_{Mnet}$ time series' (Fig. 3) demonstrate that weather conditions in 2015 impacted the soil water fluxes in the following

years: Under the dry climate in BL, hardly any drainage was observed for all soils after 2015. This indicates that the soils remained so dry during the winter period that downward water percolation or groundwater drainage was limited. The lack of water recharge during winter also affected the upward directed $Q_{Mnet}$ flux rates in the following years, which generally decreased after 2015, especially for soils from BL and Sb. The nearly unchanged $Q_{Mnet}$ values for the soils at BL after 2015 indicate that soil water saturation and dynamics is limited throughout the soil profile.


The annual net water fluxes ($Q_{Anet}$) at the bottom (in 1.5 m) of the same soils under the dry and wet climates are compared in form of scatterplots (Fig. 4). The scatterplots clearly show that fluxes were in general directed upward (i.e., negative values of $Q_{Anet}$ for soils under a dry climate in BL; positive values of $Q_{Anet}$ (i.e. drainage) were only observed for 2011 and 2014 (Fig. 4). The larger values of $Q_{Anet}$ for 2014 could be due to the lower ET after an earlier harvesting of the oat crop and a

longer bare soil period without crop transpiration. The coarser-textured soils from Dedelow showed the largest range of $Q_{Anet}$ values (from -78 mm to +164 mm) at the site with a relatively dry climate (BL) during the observation period of 2011-2017. This range could be explained by variation in soil water storage capacities between Dd soils, which depended on the thickness of the upper soil horizons that were modified by soil erosion (Herbrich et al., 2017). The long-term average values of $Q_{Anet}$ for all soils in the dry climate were negative and varied only in a small range (from -18 mm to -28 mm; see appendix

Table A1). Long term negative groundwater recharge is only possible at sites where groundwater can be replenished, for instance, by lateral subsurface water flow. Whether the $Q_{Anet}$ flux under the BL climate will continue to be negative for all soils would require a longer time series. Nevertheless, a low and even negative groundwater recharge has not only an impact on the groundwater quantity, but it will also affect the groundwater quality. In case of a small net recharge, the concentrations of solutes from agricultural fertilizers, pesticides, and those of dissolved minerals and salts in the water-filled

soil pores will become relatively high, and soil water movement still remains negligibly small. Thus under conditions of relatively small leaching rates, solutes including plant nutrients will largely be retained within the soil's root zone. Under long term conditions of net negative leaching, soils and soil horizons may accumulate carbonates (e.g., BL soil Haplic Chernozems), or if leaching is small such that the carbonates from the topsoil horizons precipitate already in the subsoil within the 1.5 m soil monoliths like in the Ccv horizons in Dd subsoil of Calcic Luvisols (see soil profile descriptions in

Herbrich and Gerke, 2017) and eventually salts.

$Q_{Anet}$ values under a relatively wet climate (in Se) were for all soils positive, indicating in general downward directed drainage fluxes (Fig. 4). The long-term average $Q_{Anet}$ values ranged between 49 to 119 mm (see appendix Table A1) depended on the soil type. The $Q_{Anet}$ value was larger for the coarser-textured soil from Dd (Fig. 4c) as compared to the other soils. For 2013 (Winter Canola crop), the $Q_{Anet}$ fluxes were negative for all finer-textured soils (i.e. Bad Lauchstädt,

Sauerbach, and Selhausen, Fig. 4a, b, d), which might be related to the deeper reaching root system of the crop canola (Breuer et al., 2003) in and a consequently larger plant water uptake in comparison to other crops. Upward directed $Q_{Anet}$





values were observed during the year 2017 for the soils from Bad Lauchstädt under winter rye crop (Fig. 4a) and during 2015 for the soils from Sauerbach under winter wheat (Fig. 4b).

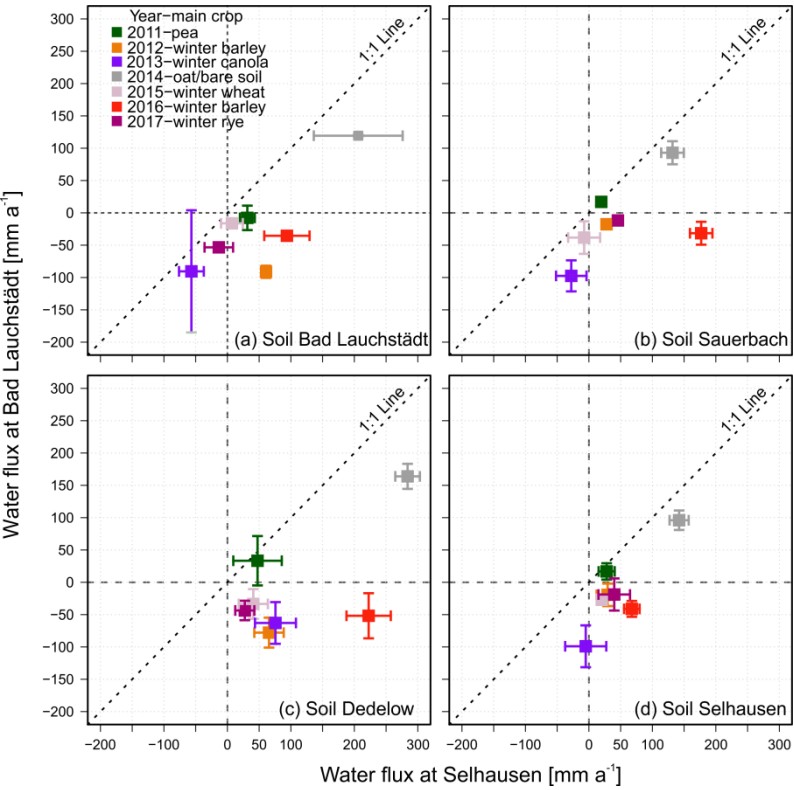

**Figure 4:** Comparison of net fluxes for the same soils at two sites: Annual observed net soil water flux at 1.5 m soil depth of soils from a) Bad Lauchstädt, b) Sauerbach, c) Dedelow, and d) Selhausen under a dry climate (Bad Lauchstädt) and wet climate (Selhausen) for the years between 2011-2017; average values (symbols) and standard deviations (error bars) for observations from the same soil.

### 3.3 Crop yield and Water Use Efficiency

The grain yields were in general larger for a dry climate at Bad Lauchstädt than for a wet climate at Selhausen except for the peas (Fig. 5a). The pea crop had in comparison to the other cereal crops a relatively short vegetation period and depends more on conditions during germination in early spring than on differences in climatic conditions in late spring and summer. For the other crops the spread of fungal pathogens under a more humid climate (Talley et al., 2002; Agam and Berliner, 2006) and frequent occurrence of dew formation (Xiao et al., 2009; Groh et al., 2018a; Brunke et al., 2019; Groh et al., 2019) could explain the generally lower yield of grain crops for soils under a wet climate in Selhausen. However, an appropriate crop management with one to three applications of fungicides during the growing season (see appendix Table A2), except for pea crop in 2011 (BL and Se) and winter rye 2017 (Se) should have prevented the spread of fungal diseases and their



impact on crop yield such that other reasons have to be considered. The yield varied for the most crops among the soil

replicates at a certain site, which can be described by the coefficient of variation (CV), below a CV value of 28%, except for pea, which showed for all soils a high value, for winter canola grown on finer-textured soils in Se (BL, Se see appendix Table A3), and for winter barley (Dd and Sb in 2012, Sb in 2016) cropped at Se. For winter canola this might be related to a higher loss of rapeseeds during manual harvesting, natural pod shattering, cleaning and threshing (Alizadeh et al., 2007; Kuai et al., 2015). The CV value of the observed yield variability between each soil type corresponds to values reported

between 5 to 27 % by Joernsgaard and Halmoe (2003) and Wallor et al. (2018). The yield of winter wheat (7.8 t ha$^{-1}$ see appendix Table A3) for the soil from BL at BL agreed well with observations on yields from a long term fertilization experiment at the BL site (Merbach and Schulz, 2013), which demonstrates the high yield potential of the soil from BL.

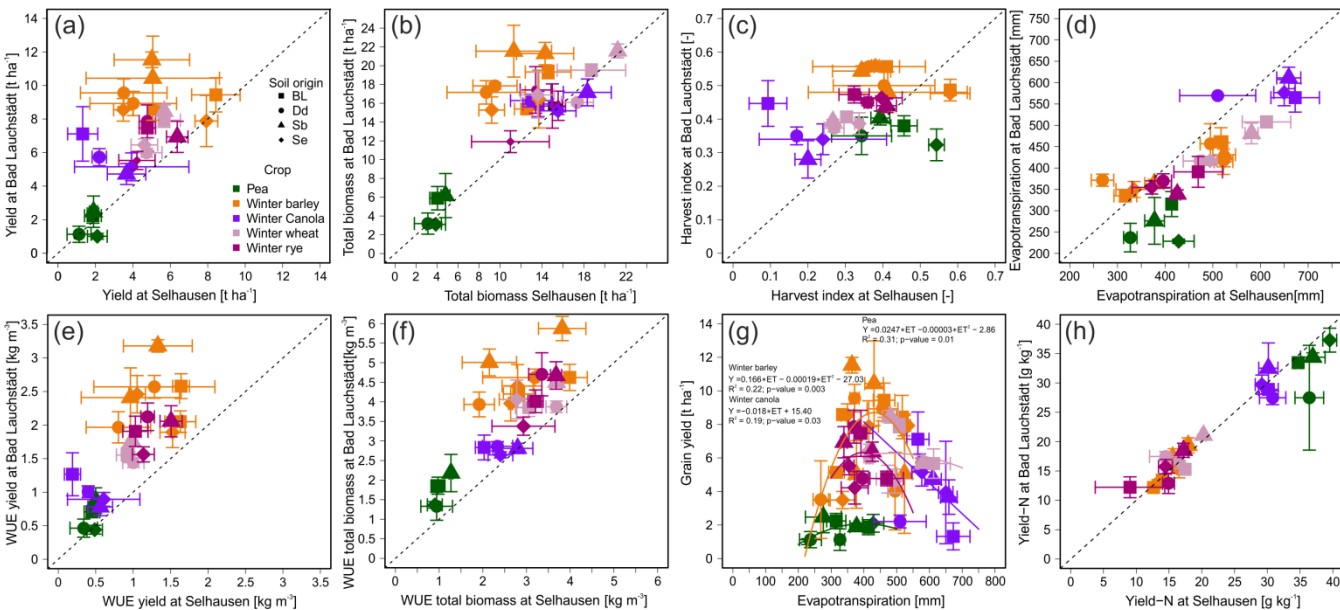

**Figure 5:** Comparison of annual crop yield- and ET-related parameters for the same soils from Bad Lauchstädt, Dedelow, Sauerbach, and Selhausen (three lysimeters each origin) at the two sites with relatively dry (Bad Lauchstädt) and wet climate (Selhausen); average values (symbols) and standard deviation (error bars) between observations from the same soils for (a) observed yield, (b) total biomass, (c) harvest index, (d) evapotranspiration, (e) water use efficiency (WUE) from yield, (f) WUE from total biomass, and (h) nitrogen (*N*) content in the grain yield, and (g) the relationship between grain yield and evapotranspiration of all soils and crops during the years 2011-
350 2017.

The scatterplot of the total biomass (Fig. 5b) shows that most crops produced relatively similar amounts of total above ground biomass at both sites with the exception of winter barley in years 2012 and 2016. The crops could probably use comparable amounts of solar radiation during the observation period (average annual radiation from 2011 to 2017, obtained

from the weather stations; BL: 1181.4 kWh m$^{-2}$ and Se: 1180 kWh m$^{-2}$). Despite a similar amount of radiation received by the crops the harvest index, which is defined as the ratio of yield to the total biomass, was found to be larger under a dry





climate than under a wet climate (Fig. 5c). This means that crops under a dry climate were more productive with respect to crop yield than under a wet climate. The crop ET (i.e., ET related to the vegetation period) was larger under the wet than under the dry climate (Fig. 5d), and the corresponding crop water use efficiency (WUE) was larger at the site with the
relatively dry (BL) as compared to the wet (Se) climate (Fig. 5e). These results demonstrated that plants were more efficient to produce yield at a site with a suboptimal water supply. The present results are in line with earlier findings from Zhang et al. (2015), who showed that the WUE reached a maximum under warm and dry and a stable minimum under warm-wet climatic conditions. Also when WUE was calculated based on the total aboveground biomass, a higher WUE was observed for the corresponding crop under a dry than under a wet climate (Fig. 5f), which demonstrated that climatic conditions were
not only beneficial for the grain yield but also for that of the straw. However, differences in fertilizer application (see appendix Table A2) with lower nitrate application in the wet site could be another reason for the differences in yield and biomass production.

The lower WUE under a wet climate might be related to a higher soil evaporation and plant canopy interception evaporation. Kunrath et al. (2018) found for the crop tall fescue that limiting nitrogen-supply conditions negatively affected WUE values
by a reduced leaf area index, leaf photosynthesis and radiation efficiency, which hence increased the ratio of soil evaporation to transpiration. Thus, we further compared the ET during periods when ET was either transpiration ($ET_T$) or evaporation ($ET_E$) dominated. The transpiration-dominated period was defined from the beginning of April, which corresponds well with the temporal increase of the monthly ET, until the time when plants reached the growth stage of ripening /maturity of their fruit or seeds about a month before harvest (see appendix Table A2). The rest of the vegetation period was defined as the
evaporation-dominated period. Evaporation was considered to be non-productive water use. The cumulative values of ET, $ET_T$ and $ET_E$ during the observation period are shown in Table 1. The differences for $ET_E$ between all soils in the dry and wet climate from 359 mm to 576 mm was larger than the differences for $ET_T$ (range: -72 mm to 199 mm). Especially the larger soil evaporation ($ET_E$) at Selhausen contributed to the lower WUE under wet climate.

The relationship between yield and ET was reported to correspond with the productivity function of crops (grain yield vs.
ET) and often assumed to be linear (Tolk and Howell, 2009; Wichelns, 2014). However, for our present data, a quadratic productivity function (Fan et al., 2018) of the winter barley and pea crops (Fig. 5g) rather than a linear one could explain the observed larger WUE of soils under a dry climate at Bad Lauchstädt. The crop winter canola could be best described by a linear productivity function with a negative slope (Fig. 5g). The other crops, winter rye and winter wheat, could neither be described by a linear nor a quadratic function. Longer time series' with more crop yield observations under different climatic
conditions would be necessary to confirm the assumed quadratic productivity function for these crops.

Grain yield quality in terms of the nitrogen content of the grains is an additional important variable to characterize the quality of legume and cereal crops (Kemanian et al., 2007). The scatterplot of the nitrogen content in the yield compares results from the same soils in the dry and wet climate (Fig. 5h). The comparison showed no effect of climatic conditions or of the fertilization on the crop grain quality. Larger deviations from the 1:1 line were only visible for the soils from Dedelow
and the crop pea under a dry climate and for soils from Bad Lauchstädt and crop winter rye under a wet climate (Fig. 5h).



Nuttall et al. (2017) remarked that heat stress during the time of flowering and higher temperatures during the post-anthesis period of crops impact grain-size and milling yield. The impact of rising temperatures and increasing $CO_2$ concentrations in the atmosphere on yield quality could affect the nutritional quality and end-use value (Asseng et al., 2019). The grain yield quality was reported to be influenced mainly by genetics, crop management, and environmental conditions (Nuttall et al.,

2017). Since in the present study, the crop management was similar and the same cultivars were used, the altered climatic conditions seemed not to affect the quality of the yield in our crossed soil-climate experiment.

**Table 1:** Average values (of 3 lysimeters each) of cumulative evapotranspiration ($\sum$ET) for the whole observation period (2011-2017), and cumulative transpiration ($\sum$ET$_T$) and evaporation ($\sum$ET$_E$) for periods dominated by evaporation (E) or transpiration (T), for soils from Bad

Lauchstädt (BL), Sauerbach (Sb), Dedelow (Dd), and Selhausen (Se) under a dry climate at BL and a wet climate at Se. The ET$_T$ values were defined from the beginning of the vegetation period (April) until ripening/maturity of the fruit or seeds; the data for $\sum$ET$_E$ comprised the values from rest of the season. The differences of the cumulative values for the same soils between the sites BL and Se are denoted by $\Delta\sum$ET, $\Delta\sum$ET$_E$ and $\Delta\sum$ET$_T$.

| Location | | Se | BL | Se | BL | Se | BL | Se | BL |
|---|---|---|---|---|---|---|---|---|---|
| Soil | | BL | BL | Sb | Sb | Dd | Dd | Se | Se |
| $\sum$ET | (mm) | 4090.1 | 3490.8 | 4121.0 | 3406.8 | 3593.9 | 3316.7 | 3985.0 | 3323.0 |
| $\sum$ET$_E$ | (mm) | 2102.5 | 1616.9 | 2110.3 | 1595.1 | 1941.7 | 1593.1 | 2228.2 | 1668.0 |
| $\sum$ET$_T$ | (mm) | 1987.5 | 1873.9 | 2010.7 | 1811.7 | 1652.1 | 1723.7 | 1756.8 | 1655.0 |
| $\Delta\sum$ET | (mm) | 599.3 | | 714.2 | | 277.1 | | 661.9 | |
| $\Delta\sum$ET$_E$ | (mm) | 485.7 | | 515.2 | | 348.7 | | 560.2 | |
| $\Delta\sum$ET$_T$ | (mm) | 113.6 | | 199.0 | | -71.5 | | 101.8 | |

## 4 Conclusion

Lysimeter data from a German-wide lysimeter network (TERENO-SOILCan), where intact soil monoliths were moved to sites with contrasting climatic conditions, were used to analyse effects of soil and climate on agricultural ecosystems in a soil-climate crossed factorial design. In the wet climate, there was a net drainage which was larger for the coarser- than for the finer-textured soils. In the dry climate, a small negative net drainage (upward flux) was obtained when observing the long-term average for the whole period 2011-2017. In the wet climate, drainage dominated for all soils. When looking at

shorter periods, negative values of monthly net fluxes observed during the summer months at both sites.

During winter months, the soil water storage (SWS) returned to a typical value and drainage occurred when this value was reached. In the dry climate, this critical SWS was not reached in two soils after the growing season of 2015 in which the SWS was strongly depleted. The resulting insufficient refilling of the soil water storage capacity after a drought suggests that the precipitation during the following winter months was not sufficient to refill the soil so that no drainage took place. This

lack of drainage had consequences for the upward water fluxes in the following growing seasons. Future studies about the impact of climate change, which in general are expected to increase the frequency and duration of droughts, on agro-



ecosystem water balances and crop development should consider the long lasting impact of droughts on the soil water balance and soil water fluxes that are carried over to following years. Results indicate that direct observation on SWS will become increasingly important in environmental climate change studies, where changing climatic conditions could affect the
SWSC. Longer term monitoring data are needed to observe effects of impacts on soil properties.

Crops were more productive in terms of grain yield and used less water under drier climatic conditions. Plant development and a higher crop water use efficiency demonstrated that less plant available soil water did not go along with a decline of grain yield, because plants used the available soil water resources under such conditions more efficiently (e.g. by reduced soil evaporation). Results revealed in contrast to our hypothesis of a linear productivity function for some crops a quadratic
productivity function and thus showed that plants can maximize their grain yield under an intermediate ET range in rainfed agriculture. However, longer time series are necessary to confirm the latter hypothesis of a quadratic productivity function of the corresponding crop. Our results suggest that despite the higher grain yield (quantity) climatic conditions seemed not to affect the quality of the yield, which might reflect a positive effect of the regional drier climatic conditions for crop production. The results of this study so far confirmed that typical soil water balance components, crop water use and
especially the soil water storage dynamics undergo a substantial change when exposed to different climatic conditions.

We could show that:

1)    The result further suggests that a new approach based on lysimeter mass data can enable the long-term monitoring of SWS changes at the pedon scale.

2)    SWS dynamics were vulnerable to droughts and led to an insufficient refilling of the soil water storage capacity.

3)    Crossed soil-climate experiments are useful to determine the impact of changing climatic conditions on the ecosystem water balances.

4)    Crop water use efficiencies were not constant and changed toward larger yields under suboptimal water supply conditions.

The results herald the need to account for potential changes in soil water storage and plant reactions due to changes in
climatic conditions and variability when trying to develop adaptation strategies in the agricultural sector.

**Data availability**

All data for the specific lysimeter and weather station (raw data) can be freely obtained from the TERENO data portal (https://teodoor.icg.kfa-juelich.de/ddp/index.jsp, lysimeter station Bad Lauchstädt and Selhausen: SE_Y_03, SE_Y_04). Climate data for the experimental station Bad Lauchstädt can be acquired upon request from Ralf Gründling. The processed
data to support the findings of this study can be acquired also from the TERENO data portal (https://hdl.handle.net/20.500.11952/butt.metadata.handle/00000010).



**Author contribution**

TP conceived the experiments. JG, JV, HHG, and TP had the idea and designed the study. JG and RG provided the data for the corresponding lysimeter stations. JG performed the data analysis and wrote the manuscript with equal contributions from all co-authors.

**Acknowledgements**

We acknowledge the support of TERENO and SOILCan, which were funded by the Helmholtz Association (HGF) and the Federal Ministry of Education and Research (BMBF). We thank the colleagues at the corresponding lysimeter station for their kind support: Sylvia Schmögner, Petra Petersohn and Ines Merbach (Bad Lauchstädt), Jörg Haase (Dedelow), Werner Küpper, Ferdinand Engels, Philipp Meulendick, Rainer Harms, and Leander Fürst (Selhausen). Thanks to Gernot Verch (Dedelow-ZALF) for helpful discussions and constructive comments.

**Competing interests**

The authors declare that they have no conflict of interest.





**Appendix data:**

**Table A1:** Average observed soil water flux at 1.5m soil depth of soils from Bad Lauchstädt (BL), Sauerbach (Sb), Dedelow (Dd), and Selhausen (Se) under a dry climate Bad Lauchstädt (BL) and wet climate Selhausen (Se).

| Origin | Transfer | 2011* | 2012 | 2013 | 2014 | 2015 | 2016 | 2017 | Average[#] |
|--------|----------|-------|------|------|------|------|------|------|---------|
|        |          | mm    | mm   | mm   | mm   | mm   | mm   | mm   | mm      |
| BL-    |          | -8    | -91  | -91  | 119  | -16  | -35  | -53  | -28     |
| Sb     | BL       | 17    | -18  | -97  | 93   | -38  | -32  | -12  | -17     |
| Dd     | BL       | 33    | -78  | -63  | 164  | -34  | -52  | -44  | -18     |
| Se     | BL       | 17    | -19  | -99  | 96   | -27  | -41  | -19  | -18     |
| BL     | Se       | 31    | 61   | -57  | 206  | 7    | 94   | -14  | 50      |
| Sb     | Se       | 19    | 28   | -28  | 132  | -8   | 177  | 46   | 58      |
| Dd     | Se       | 48    | 66   | 76   | 284  | 41   | 223  | 27   | 119     |
| Se-    |          | 28    | 30   | -5   | 142  | 20   | 68   | 40   | 49      |
| * April-December; # 2012 – 2017 | | | | | | | | | |





**Table A2:** Site management information on seasonal crop type, sowing and harvesting date, crop growth length N-fertilizer and number of fungicide applications
at Bad Lauchstädt and Selhausen. Calcium ammonium nitrate (KAS) was mainly us as *N*-fertilizer. Except for 2013 in Selhausen, where ammonium sulphate nitrate (ASS) instead of KAS was used.

| Crop | Bad Lauchstädt | | | | | Selhausen | | | | |
|------|------------------|----------------|-------------------|-----------------------------------|-------------------------------------|------------------|----------------|-------------------|-----------------------------------|-------------------------------------|
| | Sowing YY/mm/dd | Harvest YY/mm/dd | Duration [days] | N-Fertilizer (KAS) kg N ha$^{-1}$ | Number of fungicide applications | Sowing YY/mm/dd | Harvest YY/mm/dd | Duration [days] | N-Fertilizer (KAS) kg N ha$^{-1}$ | Number of fungicide applications |
| Pea | 11/ 05/04 | 11/ 08/11 | 99 | | 0 | 11/ 06/01 | 11/ 08/25 | 85 | | 0 |
| Winter Barley | 11/ 09/30 | 12/ 07/12 | 278 | 145 | 3 | 11/ 10/14 | 12/ 07/10 | 270 | 50 | § |
| Winter Canola | 12/ 08/27 | 13/ 07/23 | 330 | 210 | 2 | 12/ 09/18 | 13/ 07/25 | 310 | 130# | 1 |
| Oat | 14/ 03/13 | 14/ 06/03 | 82 | 60 | 0 | 14/ 03/05 | 14/ 06/03 | 90 | 60 | 0 |
| Winter Wheat | 14/ 10/13 | 15/ 07/28 | 288 | 60 | 2 | 14/ 10/07 | 15/ 07/21 | 279 | 90 | 2 |
| Winter Barley | 15/ 09/ 22 | 16/ 06/30 | 282 | 100 | 3 | 15/ 10/07 | 16/ 07/08 | 275 | 80 | 2 |
| Winter Rye | 16/ 10/05 | 17/ 07/17 | 285 | 100 | 2 | 16/ 10/11 | 17/ 07/21 | 283 | 78 | 0 |

# ASS; § no data available





**Table A3:** Average and coefficient of variation (CV) of yield for soil from Bad Lauchstädt (BL), Dedelow (Dd), Sauerbach (Sb), Selhausen (Se) under dry (BL)
and wet climate (Se). Origin describes the location, where the soil was taken from and the location where the soil was transferred to, where the soil was transferred to. The value in the brackets describes the variability of yield for each soil type (standard deviation from three replicates).

| Test site | | 2011 | | 2012 | | 2013 | | 2014 | 2015 | | 2016 | | 2017 | |
|---|---|---|---|---|---|---|---|---|---|---|---|---|---|---|
| Origin | Transfer | Pea | | Winter barley | | Winter canola | | Oat | Winter wheat | | Winter barley | | Winter rye | |
| | | t ha$^{-1}$ | CV % | t ha$^{-1}$ | CV % | t ha$^{-1}$ | CV % | t ha$^{-1}$ | t ha$^{-1}$ | CV % | t ha$^{-1}$ | CV % | t ha$^{-1}$ | CV % |
| BL | | 2.20 (±0.44) | 20 | 9.30 (±1.01) | 11 | 7.01 (±1.60) | 23 | # | 7.81 (±0.42) | 5 | 8.46 (±0.23) | 3 | 7.35 (±1.35) | 18 |
| Dd | BL | 1.11 (±0.48) | 43 | 8.79 (±0.71) | 8 | 5.64 (±0.50) | 9 | # | 5.90 (±0.26) | 4 | 9.42 (±0.81) | 9 | 7.74 (±0.97) | 13 |
| Sb | BL | 2.44 (±0.92) | 38 | 10.28 (±2.47) | 24 | 4.65 (±0.61) | 13 | # | 8.33 (±0.37) | 4 | 11.36 (±0.46) | 4 | 6.83 (±0.92) | 13 |
| Se | BL | 0.99 (±0.17) | 17 | 7.76 (±1.50) | 19 | 5.07 (±0.82) | 16 | # | 6.36 (±0.36) | 6 | 8.42 (±0.66) | 8 | 5.45 (±0.53) | 10 |
| BL | Se | 1.87 (±0.45) | 24 | 8.43 (±1.29) | 15 | 1.32 (±0.80) | 62 | # | 5.66 (±0.88) | 16 | 5.14 (±0.05) | 1 | 4.77 (±0.44) | 9 |
| Dd | Se | 1.13 (±0.64) | 57 | 4.02 (±2.31) | 57 | 2.21 (±0.38) | 17 | # | 4.73 (±0.83) | 18 | 3.51 (±2.31) | 1 | 4.77 (±0.44) | 9 |
| Sb | Se | 1.90 (±0.22) | 12 | 5.06 (±3.56) | 70 | 3.66 (±1.03) | 28 | # | 5.67 (±0.14) | 2 | 5.01 (±2.02) | 40 | 6.37 (±0.57) | 9 |
| Se- | | 2.10 (±0.53) | 25 | 7.93 (±0.61) | 8 | 3.94 (±3.05) | 77 | # | 4.61 (±0.71) | 15 | 3.49 (±0.51) | 15 | 4.22 (±0.97) | 23 |

\# Crop was not harvested but biomass was cut and removed in June. Manually tilled so that soil was bare fallow during summer



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
