# Peer review of "Responses of soil water storage and crop water use efficiency to changing climatic conditions: A lysimeter-based space-for-time approach"

_Hydrology and Earth System Sciences, 2019_

## Referee Comment (RC1) · Anonymous Referee #1 · 27 Sep 2019

The manuscript by Groh et al. aims primarily at evaluating if, and to what extent, changes in local climatic conditions among some German sites affect crop production and water use efficiency. The evaluation is carried out through a "space-for-time" (SFT) framework by moving soil monoliths contained in lysimeters in two locations subjected to different aridity index. Among the various outcomes of this study, the authors claim that a more efficient crop water use occurs under less optimal soil moisture conditions. The text reads well and is properly organized, although some parts are too wordy or seem going astray in describing the moving of the lysimeters. Figures and tables are satisfactory, but I suggest that the readability of Figs. 2 and 3 should be improved. As I will specify below, I have some concerns about the approach and modeling tool used,

and the discussion of some results. Therefore, while the topic is of current interest for the HESS readership, this paper should not be accepted in its present form, requires major revisions or should be rejected altogether.

1) About the SFT approach The authors employ the SFT approach in the context of moving the lysimeters from one location to another in Germany. SFT is not new, actually, and mostly used in Ecology, but some examples can be seen in the hydrologic literature (e.g. Scanlon et al., AWR 28:291-302; Troch et al., HESS 17:2209-2217). However, the way the authors have employed this approach does not seem to follow a standard (I guess), and therefore I think that an evaluation is required to test its soundness in the context of the submitted paper. The "long-term" concept exerts a key role when applying the SFT approach, but in this study only thirty years of weather data are exploited (just a minimum from a meteorological viewpoint) and then only six years are considered for the analysis (from 2012 to 2017). In view of this, I suggest that questions about "climate change" should be left out of this paper, whereas at least the authors might deal with possible changes, if any, in weather seasonality (e.g. a prolonged dry period or wet period, namely anomalies with respect to what observed during the 30 years of records). If longer time series of weather data were available (but 30 years could be used just like a threshold), plots of "Standardized Precipitation Index, SPI" or "Standardized Precipitation Evapotranspiration Index, SPEI" would definitely help. By the way, Walter and Lieth's climate diagrams for only six years is not a good practice.

2) About the modeling tool My view is that the topic coved in the paper is addressed more effectively if one looks at the derivative of the system dynamics and not at its integral behavior. In order to translate my comment in modeling terms, what I am suggesting is the use of a Richards-based model instead of the bucketing type approach expressed by Eq.(1). Giving a look at the paper by Pütz et al. (2016), I see that the lysimeters are fully equipped with soil water content and potential sensors, together with other sensing devices. Therefore, I am wondering why the authors did not exploit the potential of this information to use the Richards equation.

3) Concerns about determining ET My comment in this point 3) is linked somehow to the previous point 2). The use of ET, instead of making the partitioning of this variable in actual evaporation (Ea) and actual transpiration (Ta), can be something that may strongly limit the understanding of the ongoing processes and might yield erroneous outcomes. The use of the bucket model of Eq.(1) does not account for the important aspect of evaluating the possible onset of stress conditions in the crops and hence the computations of water use efficiency. The concept of "available water" or "readily available water" (as implied by Eq.(1), if I understood well) is definitely not adequate for the objectives of this paper. The plant can be under stress conditions due to the atmospheric demand even if a good amount of soil water is in the soil profile. Water transport resistances into the plant also play a key role. Moreover, what about possible physiological reactions of the vegetation when moving the lysimeter from one location to another? Did the authors check this aspect? Usually, vegetation shows some sort of resilience to its moving, at least during the initial stage of this moving. Can the authors comment on those points?

---

## Referee Comment (RC2) · Anonymous Referee #2 · 1 Oct 2019

The manuscript by Groh et al. presents results from the lysimeter network SOILCan. The focus of this study is on the effects of different weather and soil texture conditions on crop water use efficiency using a space-for-time approach. Hereby, weighable lysimeters with soils from four sites were moved and monitored at two of the sites with a drier and wetter climate, respectively. Instead of assessing changes in soil water storage as a residual of the water balance components the changes in lysimeter weights were used to avoid an accumulation of errors. One of the main outcomes was that the water-use-efficiency was improved (due to lower evaporation loss from soils) under drier soil moisture conditions not following a linear function. Further, the effects of drought were still visible in the following season and even beyond that especially

on finer-textured soils. Overall, the manuscript reads very well with a logical structure. The manuscript deals with the very relevant topic of changing climate conditions on agricultural productivity. The combination of weighable lysimeters in a space-for-time approach investigating four different soils with data over seven years provides valuable and interesting insights on how crop production may be affected. One of the strengths of this MS is that the authors present a comprehensive data set covering a seven-year period. The measurement data can be used for model development, calibration and validation. I recommend that the authors present such a model study in a follow-up paper. I recommend the acceptance of the manuscript upon minor revisions.

Specific comments M&M section Information about the soil texture of all four soil would be helpful as it later becomes an important part in the discussion (coarser vs finer textured soils) Figure 2 Please improve readability. Tick mark labels are very small L24 & L123 'monitored from April 2011 until December 2018' versus 'lysimeter data from April 2011 until December 2017' Please clarify. L244-264 Could this be related to a higher infiltration capacity of the coarser textured soil allowing for a more rapid recharge? It would be interesting if the authors made any observations on silting, cracking etc. of the soil surfaces especially of the finer-textured soils which might explain deficiencies in soil water recharge. L410 '. . .net fluxes were observed. . .'

---

## Author Comment (AC2) · 13 Nov 2019

Response to comments by Anonymous Referee #1:

General Comments:

The manuscript by Groh et al. aims primarily at evaluating if, and to what extent, changes in local climatic conditions among some German sites affect crop production and water use efficiency. The evaluation is carried out through a "space-for-time" (SFT) framework by moving soil monoliths contained in lysimeters in two locations subjected to different aridity index. Among the various outcomes of this study, the authors claim

that a more efficient crop water use occurs under less optimal soil moisture conditions. The text reads well and is properly organized, although some parts are too wordy or seem going astray in describing the moving of the lysimeters. Figures and tables are satisfactory, but I suggest that the readability of Figs. 2 and 3 should be improved. As I will specify below, I have some concerns about the approach and modeling tool used, and the discussion of some results. Therefore, while the topic is of current interest for the HESS readership, this paper should not be accepted in its present form, requires major revisions or should be rejected altogether.

Response: The authors thank Referee #1 for reviewing our paper. We will improve the readability of the text and figures (Figs. 2 and 3, tick mark labels) in the revised manuscript. We will also include a clearer description of the transfer of lysimeter and if necessary, a shortening of the text. The main concerns of Referee #1 on our manuscript refer to the space-for-time approach in the set-up and the used modeling approach. We think that our approach can be well justified and hope that our answers to the specific questions are convincing. To the best of our knowledge, we provide for the first time observations on water fluxes, crop yield, biomass, N-content of yield for different soil types, but each under different climatic conditions (wet & dry) following a modified space-for-time substitution.

Specific Comments:

1a) About the SFT approach. The authors employ the SFT approach in the context of moving the lysimeters from one location to another in Germany. SFT is not new, actually, and mostly used in Ecology, but some examples can be seen in the hydrologic literature (e.g. Scanlon et al., AWR 28:291-302; Troch et al., HESS 17:2209-2217). However, the way the authors have employed this approach does not seem to follow a standard (I guess), and therefore I think that an evaluation is required to test its soundness in the context of the submitted paper.

Response: The reviewer correctly notes that we did not follow the standard SFT sub-

stitution as used in ecological e.g. Pickett (1989), Blois et al. (2013), and Wogan and Wang (2018) or hydrological studies e.g. Scanlon et al. (2005) and Troch et al. (2013). Typically such SFT studies assume that spatial and temporal variations are equivalent (Pickett, 1989). By translocating soils from one test site to another (please note, that we are also carrying out lysimeter trials at the original sites, see Pütz et al. (2016)), we actually account in comparison to the standard SFT approach for unsuspected effects from the past. This means that we exclude from our observations variability effects caused by past differences in local events (disturbance), pedogenesis, or site management. We will reformulate the corresponding paragraph. The spatial transfer of intact soil monoliths in the lysimeters followed an assumed direction of climatic changes of increased temperature and precipitation. The transfer of soils between the research stations imitates a change in climatic conditions and compares for identical soils the effects of different climatic conditions on crop yield and soil water fluxes with those at the original location. By transferring lysimeters between stations, the "climatic shift" is abrupt such that we are not able to follow the gradual changes of the soil ecosystem over time as suggested in standard SFT approaches. Instead, crop yield and fluxes for the same soil under different climatic conditions are compared. We will clarify that the focus is on the comparison based on a data set covering a seven-year period and not on the modified SFT approach.

1b) The "long-term" concept exerts a key role when applying the SFT approach, but in this study only thirty years of weather data are exploited (just a minimum from a meteorological viewpoint) and then only six years are considered for the analysis (from 2012 to 2017). In view of this, I suggest that questions about "climate change" should be left out of this paper, whereas at least the authors might deal with possible changes, if any, in weather seasonality (e.g. a prolonged dry period or wet period, namely anomalies with respect to what observed during the 30 years of records).

Response: We agree that the available observation period is too short for questions about climate change and we clarify this within the revised manuscript and refer it
to change in climatic conditions or weather seasonality. However, we will also note that moving soils in regions with different climatic conditions gives us a perspective to evaluate the impact of changing climatic conditions and the longer the observation period the better. TERENO-SOILCan is an ongoing project to monitor the soil-plant-atmosphere-continuum. Considering that lysimeter operations are expensive, require relatively high maintenance, such that sustainable quality lysimeter data are still limited.

1c) If longer time series of weather data were available (but 30 years could be used just like a threshold), plots of "Standardized Precipitation Index, SPI" or "Standardized Precipitation Evapotranspiration Index, SPEI" would definitely help. By the way, Walter and Lieth's climate diagrams for only six years is not a good practice.

Response: The SPI or SPEI would help to quantify the extent of the drought in 2015. However, several studies across Europe, including our region, have shown the extent of the drought in 2015. Thus we don't think that plots of SPI or SPEI would add much information to our investigation. We used the Walter and Lieth's diagram to compare between the climate conditions at the two sites. Although 6-7 years is a relatively short time, the diagram shows relative differences between both sites during the observation period. Nevertheless, we will add two additional subplots to Fig. 1 to the revised manuscript that describe the longer term climate conditions (1988-2017) according to Walter and Lieth.

2) About the modeling tool. My view is that the topic coved in the paper is addressed more effectively if one looks at the derivative of the system dynamics and not at its integral behavior. In order to translate my comment in modeling terms, what I am suggesting is the use of a Richards-based model instead of the bucketing type approach expressed by Eq.(1). Giving a look at the paper by Pütz et al. (2016), I see that the lysimeters are fully equipped with soil water content and potential sensors, together with other sensing devices. Therefore, I am wondering why the authors did not exploit the potential of this information to use the Richards equation.

Response: We don't agree with the Referee #1 in this point and we think that this is a misunderstanding of our intention of the manuscript. We actually provide a first comprehensive data set covering a nearly seven-year period, which can be used in a next step to model the soil-plant atmosphere system. Equation 1 describes how we obtain the soil water storage change directly from the raw data and the evapotranspiration was obtained using the water balance (Equation 4). We do not consider this as a modeling tool because it does not require any assumptions on soil hydraulic properties as would be the case when using Richards' equation. It should be noted, that although a considerable amount of instruments was installed in the lysimeters, a considerable uncertainty is involved when trying to derive hydraulic properties from these measurements as previously reported by Groh et al. (2018). For the questions addressed in this paper we are directly using the measured data to calculate the water balance, thereby avoiding any uncertainties that would be introduced by model assumptions.

3) Concerns about determining ET My comment in this point 3) is linked somehow to the previous point 2). The use of ET, instead of making the partitioning of this variable in actual evaporation (Ea) and actual transpiration (Ta), can be something that may strongly limit the understanding of the ongoing processes and might yield erroneous outcomes. The use of the bucket model of Eq.(1) does not account for the important aspect of evaluating the possible onset of stress conditions in the crops and hence the computations of water use efficiency. The concept of "available water" or "readily available water" (as implied by Eq.(1), if I understood well) is definitely not adequate for the objectives of this paper. The plant can be under stress conditions due to the atmospheric demand even if a good amount of soil water is in the soil profile. Water transport resistances into the plant also play a key role. Moreover, what about possible physiological reactions of the vegetation when moving the lysimeter from one location to another? Did the authors check this aspect? Usually, vegetation shows some sort of resilience to its moving, at least during the initial stage of this moving. Can the authors comment on those points?

Response: We did not mention that we used the concept of "available water" or "readily available water". But we agree that the partitioning between E and T would be helpful to clarify further findings from our investigation, but this was beyond the scope of our study. The components E and T cannot be separated by Equation 1, which is used here only to determine the changes in soil water storage from the lysimeter mass data. We are aware that there are different methods to determine water use efficiency (e.g. use of T instead of ET), but we think that E is always related to the crop specific development and management and hence represents a kind of "cropping system" water use efficiency rather than plant water use efficiency. Although we did not monitor the crop stress status, the yield data provided an indication of the cumulative stress that was experienced by the crop. The Referee #1 mentioned possible physiological reactions of the vegetation when moving the lysimeter from one to another location. Please note that the soil monoliths were bare during the transfer and we used annual crops. The crop rotation was identical at both sites.

References:

Blois, J.L., Williams, J.W., Fitzpatrick, M.C., Jackson, S.T., Ferrier, S., 2013. Space can substitute for time in predicting climate-change effects on biodiversity. Proceedings of the National Academy of Sciences 110, 9374-9379. Groh, J., Stumpp, C., Lücke, A., Pütz, T., Vanderborght, J., Vereecken, H., 2018. Inverse Estimation of Soil Hydraulic and Transport Parameters of Layered Soils from Water Stable Isotope and Lysimeter Data. Vadose Zone Journal 17. Pickett, S.T.A., 1989. Space-for-Time Substitution as an Alternative to Long-Term Studies. In: Likens, G.E. (Ed.), Long-Term Studies in Ecology: Approaches and Alternatives. Springer New York, New York, NY, pp. 110-135. Pütz, T., Kiese, R., Wollschläger, U., Groh, J., Rupp, H., Zacharias, S., Priesack, E., Gerke, H.H., Gasche, R., Bens, O., Borg, E., Baessler, C., Kaiser, K., Herbrich, M., Munch, J.-C., Sommer, M., Vogel, H.-J., Vanderborght, J., Vereecken, H., 2016. TERENO-SOILCan: a lysimeter-network in Germany observing soil processes and plant diversity influenced by climate change. Environmental Earth Sciences 75, 1-14. Scanlon, T.M., Caylor, K.K., Manfreda, S., Levin, S.A., Rodriguez-Iturbe, I., 2005. Dynamic response of grass cover to rainfall variability: implications for the function and persistence of savanna ecosystems. Advances in Water Resources 28, 291-302. Troch, P.A., Carrillo, G., Sivapalan, M., Wagener, T., Sawicz, K., 2013. Climate-vegetation-soil interactions and long-term hydrologic partitioning: signatures of catchment co-evolution. Hydrol. Earth Syst. Sci. 17, 2209-2217. Wogan, G.O.U., Wang, I.J., 2018. The value of space-for-time substitution for studying fine-scale microevolutionary processes. Ecography 41, 1456-1468.

---

## Author Response (AR1)

Response to comments by Anonymous Referee #1:

**General Comments:**

The manuscript by Groh et al. aims primarily at evaluating if, and to what extent, changes in local climatic conditions among some German sites affect crop production and water use efficiency. The evaluation is carried out through a "space-for-time" (SFT) framework by moving soil monoliths contained in lysimeters in two locations subjected to different aridity index. Among the various outcomes of this study, the authors claim that a more efficient crop water use occurs under less optimal soil moisture conditions. The text reads well and is properly organized, although some parts are too wordy or seem going astray in describing the moving of the lysimeters. Figures and tables are satisfactory, but I suggest that the readability of Figs. 2 and 3 should be improved. As I will specify below, I have some concerns about the approach and modeling tool used, and the discussion of some results. Therefore, while the topic is of current interest for the HESS readership, this paper should not be accepted in its present form, requires major revisions or should be rejected altogether.

**Response:** The authors thank Referee #1 for reviewing our paper.

We improved the readability of the text and figures (see line 281 Fig. 2 and line 305 Fig. 3, tick mark labels) in the revised manuscript.

We included a clearer description of the transfer of lysimeter and if necessary, a shortening of the text (see Line 135 until 150).

The main concerns of Referee #1 on our manuscript refer to the space-for-time approach in the set-up and the used modeling approach. We think that our approach can be well justified and hope that our answers to the specific questions are convincing. To the best of our knowledge, we provide for the first time observations on water fluxes, crop yield, biomass, N-content of yield for different soil types, but each under different climatic conditions (wet & dry) following a modified space-for-time substitution.

Specific Comments:

1a) About the SFT approach. The authors employ the SFT approach in the context of moving the lysimeters from one location to another in Germany. SFT is not new, actually, and mostly used in Ecology, but some examples can be seen in the hydrologic literature (e.g. Scanlon et al., AWR 28:291-302; Troch et al., HESS 17:2209-2217). However, the way the authors have employed this approach does not

seem to follow a standard (I guess), and therefore I think that an evaluation is required to test its soundness in the context of the submitted paper.

**Response:** We modified the text from Line 134 – 151 to:**

Local excavated lysimeters (i.e. intact soil monoliths) were transferred between the stations to subject them to different climate regimes so as to generate a crossed soilclimate setup according to the space for time (SFT) substitution approach. It should be noted, that we did not follow the SFT substitution as used in ecological (e.g. Pickett, 1989; Blois et al., 2013; Wogan and Wang, 2018) or hydrological studies (e.g. Scanlon et al., 2005; Troch et al., 2013). Typically such SFT studies assume that spatial and temporal variations are equivalent (Pickett, 1989). By translocating soils from one test site to another while keeping some of the lysimeters at their original site, we actually account for unsuspected effects from the past. In this way we eliminate effects caused by past local events such as disturbances, pedogenesis, or site management. This is in contrast to the standard SFT approach. The spatial transfer of intact soil monoliths in the lysimeters followed an assumed direction of climatic changes of increased temperature and precipitation. For this study, we considered all arable-land lysimeter at the central sites Bad Lauchstädt and Selhausen of the TERENO-SOILCan lysimeter network. Each central experimental site contains three replicates of soils from different locations: Bad Lauchstädt (BL; Haplic Chernozems, loess), Dedelow (Dd; Calcic Luvisols and Haplic Luvisols, glacial till), Sauerbach (Sb; Colluvic Regosols; colluvial deposits), and Selhausen (Se; Haplic Luvisols, loess) allowing to investigate the response of the corresponding soil type under different climates. The transfer of soils between the research stations imitates a change in climatic conditions and compares for identical soils the effects of different climatic conditions on crop yield and soil water fluxes with those at the original location. By transferring lysimeters between stations, the "climatic shift" is abrupt such that we are not able to follow the gradual changes of the soil ecosystem over time as suggested in standard SFT approaches. Instead, crop yield and fluxes for the same soil under different climatic conditions are compared.

1b) The "long-term" concept exerts a key role when applying the SFT approach, but in this study only thirty years of weather data are exploited (just a minimum from a meteorological viewpoint) and then only six years are considered for the analysis (from 2012 to 2017). In view of this, I suggest that questions about "climate change" should be left out of this paper, whereas at least the authors might deal with possible changes, if any, in weather seasonality (e.g. a prolonged dry period or wet period, namely anomalies with respect to what observed during the 30 years of records).

**Response:** We agree that the available observation period is too short for questions about climate change and we clarified this within the revised manuscript and refer it

to change in climatic conditions or weather seasonality. However, we will also note that moving soils in regions with different climatic conditions gives us a perspective to evaluate the impact of changing climatic conditions and the longer the observation period the better. TERENO-SOILCan is an ongoing project to monitor the soil-plantatmosphere-continuum. Considering that lysimeter operations are expensive, require relatively high maintenance, such that sustainable quality lysimeter data are still limited.

1c) If longer time series of weather data were available (but 30 years could be used just like a threshold), plots of "Standardized Precipitation Index, SPI" or "Standardized Precipitation Evapotranspiration Index, SPEI" would definitely help. By the way, Walter and Lieth's climate diagrams for only six years is not a good practice.

**Response:** The SPI or SPEI would help to quantify the extent of the drought in 2015. However, several studies across Europe, including our region, have shown the extent of the drought in 2015. Thus we don't think that plots of SPI or SPEI would add much information to our investigation. We used the Walter and Lieth's diagram to compare between the climate conditions at the two sites. Although 6-7 years is a relatively short time, the diagram shows relative differences between both sites during the observation period. Nevertheless, we have added two additional subplots to Fig. 1 in the revised manuscript (see line 171) that describe the longer term climate conditions (1988-2017) according to Walter and Lieth.

2) About the modeling tool. My view is that the topic coved in the paper is addressed more effectively if one looks at the derivative of the system dynamics and not at its integral behavior. In order to translate my comment in modeling terms, what I am suggesting is the use of a Richards-based model instead of the bucketing type approach expressed by Eq.(1). Giving a look at the paper by Pütz et al. (2016), I see that the lysimeters are fully equipped with soil water content and potential sensors, together with other sensing devices. Therefore, I am wondering why the authors did not exploit the potential of this information to use the Richards equation.

**Response:** We don't agree with the Referee #1 in this point and we think that this is a misunderstanding of our intention of the manuscript. We actually provide a first comprehensive data set covering a nearly seven-year period, which can be used in a next step to model the soil-plant atmosphere system. Equation 1 describes how we obtain the soil water storage change directly from the raw data and the evapotranspiration was obtained using the water balance (Equation 4). We do not consider this as a modeling tool because it does not require any assumptions on soil hydraulic properties as would be the case when using Richards' equation. It should

be noted, that although a considerable amount of instruments was installed in the lysimeters, a considerable uncertainty is involved when trying to derive hydraulic properties from these measurements as previously reported by Groh *et al.* (2018). For the questions addressed in this paper we are directly using the measured data to calculate the water balance, thereby avoiding any uncertainties that would be introduced by model assumptions.

3) Concerns about determining ET My comment in this point 3) is linked somehow to the previous point 2). The use of ET, instead of making the partitioning of this variable in actual evaporation (Ea) and actual transpiration (Ta), can be something that may strongly limit the understanding of the ongoing processes and might yield erroneous outcomes. The use of the bucket model of Eq.(1) does not account for the important aspect of evaluating the possible onset of stress conditions in the crops and hence the computations of water use efficiency. The concept of "available water" or "readily available water" (as implied by Eq.(1), if I understood well) is definitely not adequate for the objectives of this paper. The plant can be under stress conditions due to the atmospheric demand even if a good amount of soil water is in the soil profile. Water transport resistances into the plant also play a key role. Moreover, what about possible physiological reactions of the vegetation when moving the lysimeter from one location to another? Did the authors check this aspect? Usually, vegetation shows some sort of resilience to its moving, at least during the initial stage of this moving. Can the authors comment on those points?

**Response:** We did not mention that we used the concept of "available water" or "readily available water". But we agree that the partitioning between E and T would be helpful to clarify further findings from our investigation, but this was beyond the scope of our study. The components E and T cannot be separated by Equation 1, which is used here only to determine the changes in soil water storage from the lysimeter mass data. We are aware that there are different methods to determine water use efficiency and discussed this in line 86 to 92 (e.g. use of T instead of ET), but we think that E is always related to the crop specific development and management and hence represents a kind of "cropping system" water use efficiency rather than plant water use efficiency at the leaf level. Although we did not monitor the crop stress status, the yield data provided an indication of the cumulative stress that was experienced by the crop. The Referee #1 mentioned possible physiological reactions of the vegetation when moving the lysimeter from one to another location. Please note that the soil monoliths were bare during the transfer and we used annual crops. The crop rotation was identical at both sites.

Groh, J., Stumpp, C., Lücke, A., Pütz, T., Vanderborght, J., Vereecken, H., 2018. Inverse Estimation of Soil Hydraulic and Transport Parameters of Layered Soils from Water Stable Isotope and Lysimeter Data. Vadose Zone Journal 17.

**Response to comments by Anonymous Referee #2**

**General Comments:**

The manuscript by Groh et al. presents results from the lysimeter network SOILCan. The focus of this study is on the effects of different weather and soil texture conditions on crop water use efficiency using a space-for-time approach. Hereby, weighable lysimeters with soils from four sites were moved and monitored at two of the sites with a drier and wetter climate, respectively. Instead of assessing changes in soil water storage as a residual of the water balance components the changes in lysimeter weights were used to avoid an accumulation of errors. One of the main outcomes was that the water-use-efficiency was improved (due to lower evaporation loss from soils) under drier soil moisture conditions not following a linear function. Further, the effects of drought were still visible in the following season and even beyond that especially on finer-textured soils. Overall, the manuscript reads very well with a logical structure. The manuscript deals with the very relevant topic of changing climate conditions on agricultural productivity. The combination of weighable lysimeters in a space-for-time approach investigating four different soils with data over seven years provides valuable and interesting insights on how crop production may be affected. One of the strengths of this MS is that the authors present a comprehensive data set covering a seven-year period. The measurement data can be used for model development, calibration and validation. I recommend that the authors present such a model study in a follow-up paper. I recommend the acceptance of the manuscript upon minor revisions.

**Response**: The authors thank Referee #2 for reviewing our paper and their positive feedback/ comments concerning our manuscript. We are currently conducting a study and use lysimeter data for the calibration of different crop models.

Specific Comments:

M&M section information about the soil texture of all four soil would be helpful as it later becomes an important part in the discussion (coarser vs finer textured soils)

**Response**: We added as suggested a soil profile description in the supplement of the revised manuscript (see Table A1; line 486).

Figure 2 Please improve readability. Tick mark labels are very small

**Response**: We changed the tick mark labels to improve the readability of Figure 2 and 3 (see line 281 and 305 in the revised manuscript)

L24 & L123 'monitored from April 2011 until December 2018' versus 'lysimeter data from April 2011 until December 2017' Please clarify.

**Response:** We changed December 2018 to December 2017 in line 23

L244-264 Could this be related to a higher infiltration capacity of the coarser textured soil allowing for a more rapid recharge? It would be interesting if the authors made any observations on silting, cracking etc. of the soil surfaces especially of the finer-textured soils which might explain deficiencies in soil water recharge.

**Response:** The infiltration capacity is dependent on the conductivity at the soil surface. Silting, which more often occur at the soil surface of fine textured soil, affects the macropore structure (destruction of soil aggregates) and reduce the infiltration. No surface runoff was observed during the observation period. Thus we don't think that the annual carry –over of soil moisture deficits are related to a different infiltration capacity of the soil.

Some qualitative observations were made during the harvest time, but the soil surface has been modified by tillage, and the topsoil organic matter content and the plant roots are counteracting silting and cracking. We included this information and discussion in the revised paper (see line 268 - 274).

L410 '. . .net fluxes were observed. . .'

**Response**: We changed "net fluxes observed" to "net fluxes were observed" in the revise manuscript (see line 438 in the revised manuscript)

**Responses of soil water storage and crop water use efficiency to changing climatic conditions: A lysimeter-based space-for-time approach**

Jannis Groh1, 2, Jan Vanderborght2, Thomas Pütz2, Hans-Jörg Vogel3, Ralf Gründling3, Holger Rupp3, Mehdi Rahmati4, Michael Sommer5, 6, Harry Vereecken2, Horst H. Gerke1

[revised manuscript text omitted]